# Fabrication of robust and cost-efficient Hoffmann-type MOF sensors for room temperature ammonia detection

Sa Wang[1,2], Yu Fu[3], Ting Wang[1,2], Wansheng Liu[1,2], Jian Wang[1,4], Peng Zhao[2], Heping Ma [3], Yao Chen [1,4], Peng Cheng [1,2,5] & Zhenjie Zhang [1,2,5] ✉

The development of fast-response sensors for detecting $NH_3$ at room temperature remains a formidable challenge. Here, to address this challenge, two highly robust Hoffmann-type metal-organic frameworks are rationally applied as the $NH_3$ sensing materials which possess ultra-high static adsorption capacity for $NH_3$, only lower than the current benchmark material. The adsorption mechanism is in-depth unveiled by dynamic adsorption and simulation studies. The assembled interdigital electrode device exhibits low detection limit (25 ppb) and short response time (5 s) at room temperature, which set a record among all electrical signal sensors. Moreover, the sensor exhibits excellent selectivity towards $NH_3$ in the presence of 13 other potential interfering gases. Prominently, the sensor can stably output signals for more than two months at room temperature and can be recovered by simply purging nitrogen at room temperature without heating. This study opens up a way for reasonably designing gas sensing materials for toxic gases.

Ammonia ($NH_3$) is one of the most essential chemicals in the world and an irreplaceable raw material in global agriculture and industry[1–3]. Meanwhile, $NH_3$ is also a colorless, irritating and corrosive gas with high toxicity[4]. Long-term exposure to $NH_3$ with a concentration greater than 50 ppm will lead to temporary blindness, pulmonary edema, and even death. Occupational Safety and Health Administration has set a limit of 300 ppm for industrial environments[5]. Therefore, the detection and sensing of $NH_3$ are of great significance for environmental protection and human health. Currently, the use of metal oxides as the affinity layer is the mainstream in commercial $NH_3$ sensors. However, drawbacks still exist for current detection devices, e.g., low selectivity and high operating temperatures. Therefore, the rational design of advanced materials for $NH_3$ sensing at ambient conditions (e.g., room temperature) with high selectivity is urgently needed.

Metal-organic frameworks (MOFs) possess tailored structures, high porosity, and customizable functionalities are considered to be a promising choice to serve as the affinity layer of gas sensors[6–9]. Firstly,

the permanent porosity and regular channels of MOFs provide feasibility for fast sensing response and low working temperature. Secondly, the tunable pore microenvironment (e.g., pore shape and size) can adjust the host-guest interaction at the molecular level, which is the potential to offer high selectivity to targeted guest molecules. More importantly, the physical adsorption/desorption with good recyclability will endow MOF-based gas sensors with long-term reliability. Therefore, MOF-based gas sensors have been gaining increasing attention in recent years[10–13]. Although significant progress has been made in gas sensing using MOFs, the development of MOFs for $NH_3$ sensing is still in its infancy. Firstly, most MOFs exhibit structural degradation after adsorbing $NH_3$ and low binding affinity to $NH_3$, hindering the application of MOFs as $NH_3$ sensors. Secondly, the mechanism of $NH_3$ sensing is not well understood, mainly due to the lack of appropriate techniques such as adsorption kinetics and molecular simulation. Thirdly, currently reported MOF materials often require additional material doping for sensing (e.g., Cu-BTC@GO[14],

[1]College of Chemistry, State Key Laboratory of Medicinal Chemical Biology, Nankai University, Tianjin 300071, China. [2]Key Laboratory of Advanced Energy Materials Chemistry, Ministry of Education, Nankai University, Tianjin 300071, China. [3]School of Chemical Engineering and Technology, Xi'an Jiaotong University, Xi'an 710049, PR China. [4]College of Pharmacy, Nankai University, Tianjin 300071, China. [5]Frontiers Science Center for New Organic Matter, Renewable energy conversion and storage center, Nankai University, Tianjin 300071, China. ✉e-mail: zhangzhenjie@nankai.edu.cn

PANI/UiO-66[15]), which in turn requires complex fabrication procedures and even high-temperature sensing conditions, complicating the sensor device. Using pure MOF materials for $NH_3$ sensing applications is still very rare and all currently reported examples suffer from expensive ligands which are not readily available (e.g., Cu-HHTP[16–18] and $Cu_3HITP_2$[19,20]), leading to increased operating costs. Hence, it is of great significance to reasonably design cost-efficient and high-performance MOF materials for $NH_3$ sensing at room temperature.

To address the above challenge, we apply the following principles to screen MOF candidates. (i) High structural robustness is required to prohibit the structure collapse in the $NH_3$ environment. (ii) In order to achieve strong $NH_3$ binding capacity, MOFs with open metal sites (OMS) and micropores are preferred; (iii) Low-cost and large-scale preparation of MOFs is a key requirement for industrial applications such as toxic gases and air filters. Thus, we delicately choose two highly robust Hofmann-type MOFs, $Ni(pyz)[Ni(CN)_4]$ and $Co(pyz)[Ni(CN)_4]$ (termed as NiNi-Pyz and CoNi-Pyz, respectively, Pyz = pyrazine), as the sensing materials for detecting $NH_3$. Although these two materials have been previously reported for application in gas separation[21], they have not yet been investigated for sensing application. Their high-performance interdigitated electrode (IDE) sensors have been successfully fabricated and used for $NH_3$ sensing (Fig. 1).

## Results

### Structural characterization

MNi-Pyz (M = Ni and Co) were prepared through the reaction $M(NO_3)_2·6H_2O$, pyrazine and $K_2[Ni(CN)_4]$ in a green synthesis fashion (water/methanol solution, room temperature). The crystal structures of MNi-Pyz were confirmed by powder X-ray diffraction (PXRD) pattern, which was consistent with that of the simulated structure, indicating the high phase purity (Supplementary Fig. 1). In MNi-Pyz, the $[Ni(CN)_4]^{2-}$ inorganic ligands serving as 4-connected planar building blocks can connect with $M^{2+}$ nodes to form 2D layers which are further linked by pyrazine ligands to generate the 3D network. There are one-dimensional (1D) square channels along the a-axis direction. The $M^{2+}$ ions with unsaturated metal sites are uniformly arranged on the walls of 1D channels (Fig. 2). The riched open metal sites and ultramicrosized pores make MNi-Pyz a good candidate to capture gas molecules such as $NH_3$.

It was found that MNi-Pyz exhibited excellent structural stability. Their crystallinity and porosity were well maintained even after boiling water, RH = 85%, alkaline (pH = 13), or acid solution (pH = 1) treatment for one day (Supplementary Figs. 1, 2). $N_2$ sorption isotherms at 77 K showed typical type I reversible isotherms, indicative of the microporosity of the MOFs. The Brunauer−Emmett−Teller (BET) surface area of NiNi-Pyz and CoNi-Pyz were 488 and 585 $m^2·g^{-1}$ with the pore size distribution around 7.2 Å and 6.8 Å, respectively (Supplementary Fig. 3). Scanning electron microscopy (SEM) images revealed that the materials have regular morphology and blocky structure, indicating a

good crystalline state (Supplementary Fig. 4). The thermal stability of the material was investigated by in-situ thermogravimetric analysis (TGA) (Supplementary Fig. 5). The weight loss at about 120 °C corresponded to the removal of solvent molecules (methanol and water). Furthermore, in-situ variable-temperature PXRD results verified the MOFs could be stable up to 300 °C under air atmosphere (Fig. 3a, b), highlighting their high thermal stability. At present, most MOFs are synthesized via solvothermal reactions. Notably, MNi-Pyz MOFs can be synthesized quickly at room temperature with stirring for 8 min, and can also be prepared on a large scale (>10 g), endowing MNi-Pyz MOFs with industrial application potentials (Supplementary Fig. 6). Overall, the ultramicroporous structure, ultrahigh stability, and scalable synthesis make MNi-Pyz MOFs an ideal platform for gas capture and sensing applications.

### Ammonia adsorption

The adsorption curves can show the affinity behavior of the material for gas molecules, and the low-pressure strong capture ability is the basis of $NH_3$ detection. Subsequently, we explored the $NH_3$ adsorption behavior of MNi-Pyz. (Fig. 3c and Supplementary Fig. 7). It could be seen from the adsorption isotherm at 298 K that the adsorption capacity rose sharply in the low-pressure region, and then a turning point occurred at $P/P_0 = 0.21$ and 0.43 for NiNi-Pyz and CoNi-Pyz, respectively. Thereafter, the adsorption capacity continued to increase and finally reached adsorption saturation, indicating a stepwise adsorption behavior. The high adsorption capacity under low-pressure provided an experimental basis for the subsequent development of $NH_3$ sensors. The $NH_3$ uptakes of NiNi-Pyz and CoNi-Pyz reached 21.5 and 29.1 $mmol·g^{-1}$ at 298 K and 1 bar, respectively. The difference in adsorption capacity may be due to the larger BET surface area and pore volume of CoNi-Pyz than NiNi-Pyz. To further verify the ability to capture $NH_3$, the materials were placed in air for a month, and then $NH_3$ capture ability for five cycles was conducted (Fig. 3d). The results showed that their maximum uptakes of $NH_3$ were consistent in at least five cycles. We compared with other reported materials for $NH_3$ capture, including MOFs, polymers, inorganic materials, zeolite, molecular sieves, silica gel and other commercial materials (Supplementary Table 1). Notably, the adsorption capacity of MNi-Pyz was significantly higher than that of commercial materials (Amberlyst[22], 11.34 $mmol·g^{-1}$ and 13X zeolites[22], 9.3 $mmol·g^{-1}$) and famous MOFs (Cu(cyhdc)[23], 17.5 $mmol·g^{-1}$ and $Cu_2Cl_2BBTA$[24], 19.7 $mmol·g^{-1}$), and only slightly lower than the benchmark MOF (LiCl@MIL-53-$(OH)_2$[25], 33.9 $mmol·g^{-1}$). Nevertheless, taking into account the green, cheap raw materials and low energy consumption in the preparation process, MNi-Pyz can easily surpass the current benchmark MOF in terms of cost. The excellent $NH_3$ capture ability of MNi-Pyz MOFs could be due to their special pore microenvironment with enriched open metal sites and strong regional restriction effect for $NH_3$. In the real world, the competitive adsorptions of $H_2O$ and $NH_3$ often occur due to the similar

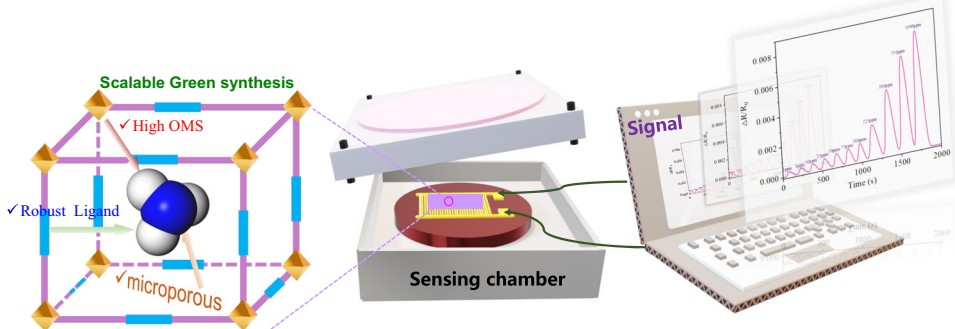

**Fig. 1 | Sensing material selection and evaluation.** Rational selection of stable microporous MOFs for $NH_3$ sensing.

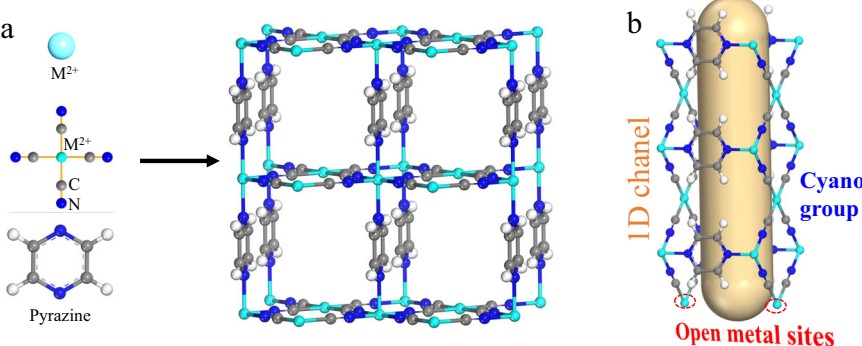

**Fig. 2 | Structure diagram of MNi-Pyz.** The open metal position of metal $M^{2+}$ is shown in cyan, the carbon atom is grey, the nitrogen atom is blue, and the hydrogen atom is white.

**Fig. 3 | Characterization and adsorption of MNi-Pyz. a, b** In-situ variable-temperature powder diffraction pattern of NiNi-Pyz and CoNi-Pyz in air atmosphere. **c** Adsorption isotherms for $NH_3$ at 298 K. **d** Five-cycle of $NH_3$ capacities at 298 K even after placing MNi-Pyz in air for a month.

molecular properties of $H_2O$ and $NH_3$. Adsorbents with high $NH_3$ uptake but relatively low uptake of water are desirable. Therefore, we tested the water vapor adsorption isotherms for MNi-Pyz. When the relative humidity (RH) rises to 60%, the water vapor adsorption capacities of NiNi-Pyz and CoNi-Pyz are 1.16 and 1.88 mmol·g$^{-1}$, respectively. When the humidity rises to 90% RH, MNi-Pyz still maintains a low water vapor adsorption capacity of 1.7 mmol·g$^{-1}$

(Supplementary Fig. 8), far below the $NH_3$ adsorption capacity, indicating efficient $NH_3$ sensing performance in the presence of water vapor.

**Dynamic test**

The rapid adsorption of $NH_3$ is related to the rapid response behavior of the sensing material, we conducted dynamic adsorption tests of $NH_3$

on MNi·Pyz (Fig. 4 and Supplementary Fig. 9). The experimental results showed that NiNi·Pyz reached 80% saturation after 20 s for the first stage of preferential adsorption, and the maximum adsorption rate is as high as 1.67 mmol·(g·s)$^{-1}$. Similarly, for CoNi·Pyz, the time to reach 80% saturation for $NH_3$ was 50 s, and the maximum adsorption rate is as high as 1.70 mmol·(g·s)$^{-1}$. These results indicated that both NiNi·Pyz and CoNi·Pyz exhibited fast dynamic adsorption for $NH_3$. Notably, NiNi·Pyz has a higher average adsorption rate than CoNi·Pyz possibly because NiNi·Pyz had a higher density of open metal sites (NiNi·Pyz: 9.29 mmol·cm$^{-3}$ and CoNi·Pyz: 8.97 mmol·cm$^{-3}$). The fast adsorption performance in kinetics lays a solid foundation for sensing applications.

## Molecular simulation

To better understand the $NH_3$ capture behavior of MNi·Pyz, we conducted Monte Carlo simulations (GCMC) performed to investigate the interactions between materials and $NH_3$. The total adsorption field shows that $NH_3$ is stably bound inside the cavity of MNi·Pyz, due to the gas molecules being subject to the barrier of surrounding ligands (Supplementary Fig. 10). The simulation results show that NiNi·Pyz has three binding sites on $NH_3$ molecules (Fig. 5a). For the binding site I (Fig. 5b and Supplementary Fig. 11a), $NH_3$ molecules are preferentially

located in the middle of the two open nickel (II) sites in the NiNi·Pyz square channel, with a distance of 2.709 Å. A sandwich-like binding environment is constructed, which is attributed to the strong interaction between the N atom of $NH_3$ and $Ni^{2+}$. For the adsorption site II (Fig. 5c and Supplementary Fig. 11b), the hydrogen atom in $NH_3$ can combine with the nitrogen atom of $[Ni(CN)_4]^{2-}$ to generate a strong electrostatic attraction, increasing the van der Waals interaction forces between $NH_3$ and ligands. In addition, the N atom of $NH_3$ can achieve conjugation with the large π bond on the pyrazine ring to achieve dual and strong interactions. For site III (Fig. 5d and Supplementary Fig. 11c), firstly, the hydrogen atom in $NH_3$ can interact with C≡N in $[Ni(CN)_4]^{2-}$ because of hyperconjugation. More importantly, $NH_3$ molecule itself tends to form dimers[26–29]. When further adsorption occurs, $NH_3$ are close to each other due to the effect of orientation force. Finally, hydrogen atoms are inserted into the electronic cloud of nitrogen atoms, and the two $NH_3$ molecules are combined by overlapping orbits to form adsorption site III. For CoNi·Pyz materials, there are four $NH_3$ molecular interaction sites (Supplementary Fig. 12). Similar to NiNi·Pyz, $NH_3$ in site I is preferentially located in the middle of two open cobalt (II) sites in the square channel, showing a lateral binding mode, thus forming a sandwich-like binding environment. For site II, the electrostatic attraction between $NH_3$ molecule and the N atom of

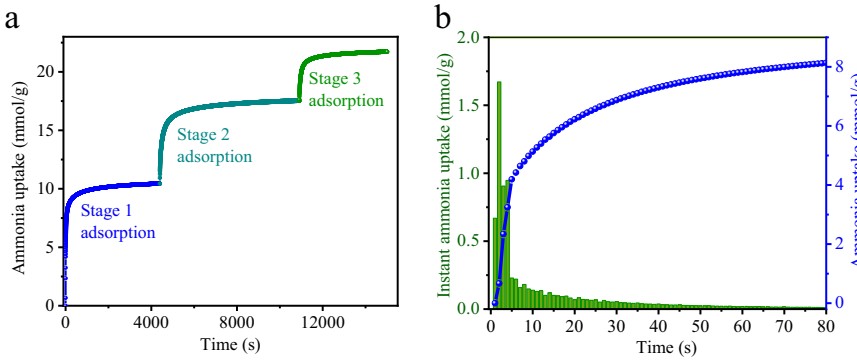

**Fig. 4 | Dynamic test for NiNi·Pyz. a** Kinetic adsorption curves. **b** Preferential stage 1 adsorption curve for NiNi·Pyz to $NH_3$ at 298 K.

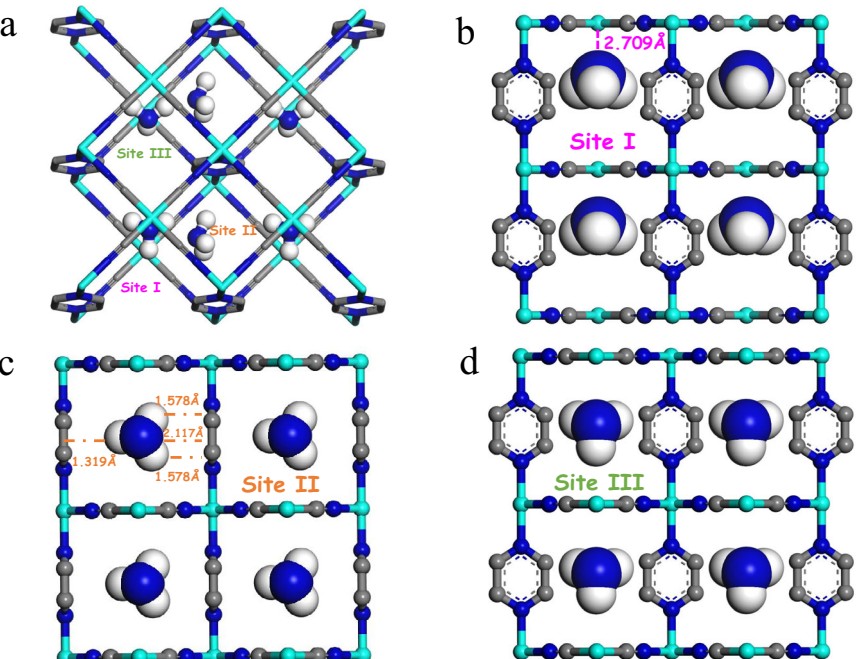

**Fig. 5 | GCMC simulated $NH_3$ adsorption field distribution of NiNi·Pyz. a** Distribution of total adsorption sites. **b–d** Binding sites for site I, site II and site III, respectively.

$[Ni(CN)_4]^{2-}$ and the conjugation with a large $\pi$ bond on the pyrazine ring realize strong interaction. For site III, when further adsorption occurs, the N atom has a partial negative charge, while the H atom has a partial positive charge. Due to the effect of orientation force, a dimer is formed to act on site III. Besides, CoNi-Pyz has an additional adsorption site IV with a weak force. In this site, $NH_3$ molecules mainly exist in the middle of the pore, which may be due to the larger surface area of CoNi-Pyz, so it can accommodate more $NH_3$ molecules. Furthermore, the interaction energy between $NH_3$ and MNi-Pyz was calculated using the Density functional theory (DFT) method (Supplementary Table 2). For site I with strong adsorption, the energy of NiNi-Pyz is higher than that of CoNi-Pyz, which is consistent with the experimental results (faster adsorption for NiNi-Pyz under low $NH_3$ pressure). The high total adsorption energy also implies that CoNi-Pyz has a higher $NH_3$ capture capacity than NiNi-Pyz. The independent action and mutual cooperation of all binding sites reveal the key points and mechanism of strong $NH_3$ capture and detection performance of MNi-Pyz. Simultaneously, we conducted simulations to analyze the adsorption sites and energies of MNi-Pyz materials towards three common air gases ($N_2$, $O_2$, and $H_2O$), as well as three typical volatile organic compounds (acetone, DMF, and ethanol). The findings demonstrated that MNi-Pyz materials exhibit higher adsorption energy for $NH_3$, and relatively lower binding energy for other gas molecules, which further confirms the specific recognition capability towards $NH_3$ of MNi-Pyz materials (Supplementary Figs. 13, 14).

**Ammonia detection.** In order to avoid the toxicity and corrosivity of $NH_3$ to human health, sensing $NH_3$ in low concentrations is of great importance. The adsorption curve and kinetic rapid adsorption in the low-pressure region offer MNi-Pyz high potentials for $NH_3$ sensing. Prior to sensing applications, the stability of MNi-Pyz in $NH_3$ was pre-evaluated. It can be found that the structure of MNi-Pyz remained unchanged when the materials were placed at 100–1000 ppm (1000 ppm is the highest concentration currently used for sensing) after one day (Supplementary Fig. 15). To further verify material

stability, we placed MNi-Pyz in an environment containing 1000 ppm $NH_3$ for up to one month, and the materials still maintained their crystalline structures (Supplementary Figs. 16, 17). The excellent structural stability of MNi-Pyz in the presence of $NH_3$ provides a strong guarantee for $NH_3$ sensor applications.

A homemade IDEs setup was used for gas sensing measurements (Supplementary Fig. 18). In this study, MOFs with particle size of ~200 nm (Supplementary Fig. 19) were uniformly coated on the inter-digital electrode (Supplementary Fig. 20) and put into a stainless-steel chamber with a total volume of 220 $cm^3$. The chamber was equipped with a device connected to temperature control and a mass flow controller for gas supply. The test results showed that MNi-Pyz sensors had a good linear response in a wide range of 1–1000 ppm at room temperature. According to the root mean square deviation[30,31], the NiNi-Pyz sensor detection limit was calculated as 25 ppb (Fig. 6a). NiNi-Pyz achieved rapid response and recovery to $NH_3$, with a response time of about 5 s and recovery time of about 55 s in $N_2$ atmosphere (Fig. 6b). Similarly, for CoNi-Pyz sensor, the $NH_3$ detection limit was 97 ppb, the response time was estimated to be around 20 s and the recovery time was 45 s at $N_2$ atmosphere (Supplementary Fig. 21 and Table 3). Based on the above results, to better understand the sensing performance of MNi-Pyz, we compared the reported MOFs as the main affinity layer for $NH_3$ sensor (Supplementary Table 4). It can be found that the sensing detection limit of MNi-Pyz is only lower than that of the current benchmark, Cu-HHTP 3D films[14], but higher than that of classical materials such as Cu-HHTP[15], $Cu_3HITP_2$[17], $Ni_3(HHTP)_2$[18], Cu-BHT film[32] and NiPc-Ni[33]. Notably, the sensing response time of MNi-Pyz surpasses all reported MOF materials. On a wider scale, compared to other excellent $NH_3$-sensing electrical signal materials reported so far (Supplementary Table 5), MNi-Pyz sensing performance is better than that of famous materials such as PEDOT[34], $MoS_2/Co_3O_4$[35], $MoS_2$ thin fIlms[36], 2D $Ti_3C_2Tx$[37] et al.

Additionally, one of the most important evaluation parameters of the sensor includes stability and reproducibility. In this regard, we conducted a cyclic stability test on MNi-Pyz of 1 - 1000 ppm $NH_3$ at

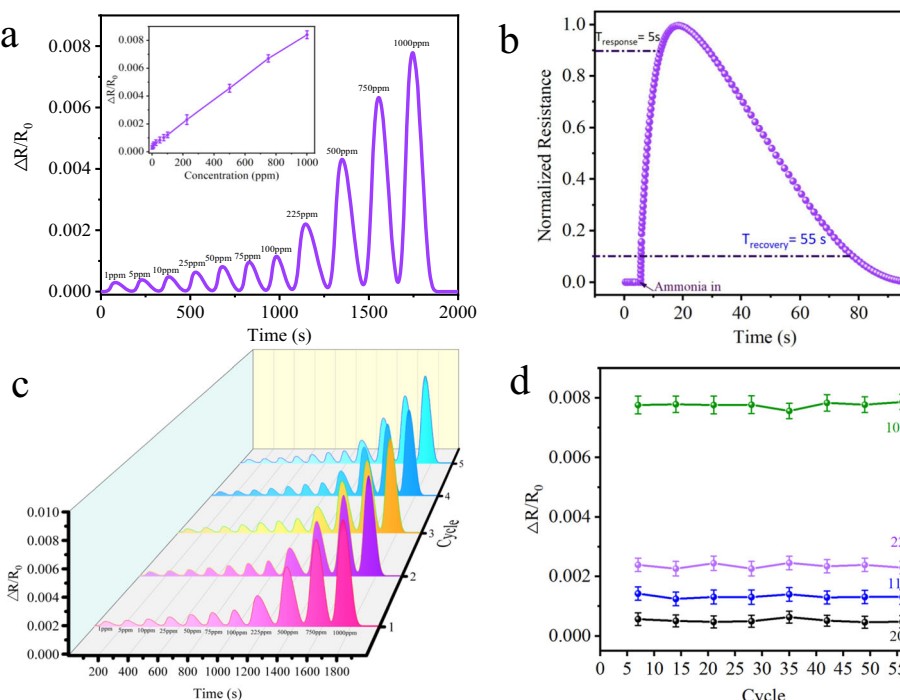

**Fig. 6 | Sensing performance for $NH_3$. a** Detection of $NH_3$ in different ranges of ppm concentrations (1–1000 ppm) using NiNi-Pyz sensor. Insets: Linear response in the corresponding range with error bars depicted. Data show means ± SD ($n$ = 5 replicates). **b** Response−recovery time curves of NiNi-Pyz sensor. **c** Performance of five cycles at 1–1000 ppm of NiNi-Pyz sensor. **d** Stability of $NH_3$ detection for NiNi-Pyz at 20, 115, 225 and 1000 ppm. Data show means ± SD ($n$ = 5 replicates).

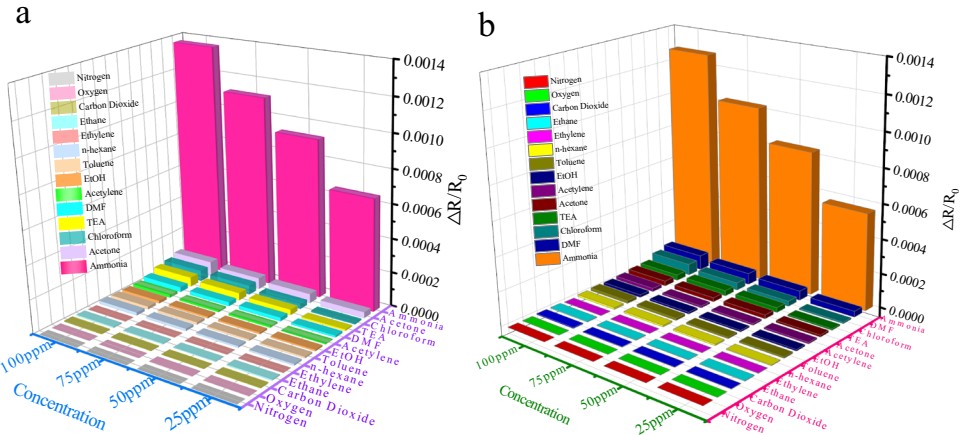

**Fig. 7 | Specific sensing evaluation.** Response toward NH$_3$ and interfering gases for **a** NiNi-Pyz. **b** CoNi-Pyz.

298 K (Fig. 6c and Supplementary Fig. 22a). Through the experiment, it proved that MNi-Pyz could conduct at least five times of NH$_3$ cycle detection, with stable and uniform detection levels and excellent stability. Further, we conducted a two-month cycle test on the sensor under the conditions of 20, 115, 225, and 1000 ppm NH$_3$ (Fig. 6d and Supplementary Fig. 22b). The test results further verify the high stability and repeatability of MNi-Pyz sensors in different cycle ranges. Subsequently, we tested the cross-sensitivity of the MNi-Pyz sensor to 13 potential interfering gases, such as volatile organic gases commonly found in the air and some reducing gases, to ensure the selectivity of the sensor[38–41]. Figure 7 showed MNi-Pyz sensors had selectivity >10 (S = Response (NH$_3$)-Response(N$_2$)/Response(N$_2$)) towards interference gases, indicating their excellent selectivity. Furthermore, to verify the practicality of the sensor, we explored the sensor signal in a real air atmosphere (RT = 298 K, RH ≈ 35–55%). From Supplementary Figs. 23, 24, it can be seen that the sensor signal in an air environment is slightly lower than that in a nitrogen environment. Meaningfully, it still has good cyclic stability and excellent selectivity, which can be further used in the real environment.

To assess the impact of humidity on sensor performance, we conducted experiments to analyze the NH$_3$ sensing behavior of the MNi-Pyz sensor under varying relative humidity (RH) conditions of 12, 35, 75, and 90% (Supplementary Figs. 25, 26). Our findings demonstrated that when the relative humidity remains below 40%, the response value remained unaffected by humidity, exhibiting a consistent sensing performance of over 80%. These results align with previously reported findings in the literature[31,42–45]. As humidity levels rose, the affinity for H$_2$O molecules increased, intensifying the competitive adsorption between H$_2$O and NH$_3$. Additionally, elevated humidity levels promoted the formation of a water film on the sensor surface, which altered the interface characteristics between the sensor and the gas. Consequently, this change in interface characteristics weakened the interaction between the sensor and the target gas, resulting in reduced sensitivity[46–48]. Subsequently, we tested the cyclic sensing performance of the sensor at 100, 225, and 500 ppm NH$_3$ at 35% RH over two-month test period, indicating the high stability of the sensor (Supplementary Fig. 27). It is worth noting that most commercial sensors must use water adsorbents before use to ensure the stability of the sensing signal. Rationally speaking, MNi-Pyz materials have demonstrated high potential for practical NH$_3$ sensing applications.

In summary, two Hoffmann-type MOFs, Ni(pyz)[Ni(CN)$_4$] and Co(pyz)[Ni(CN)$_4$], were rationally applied as the sensing materials for detecting NH$_3$. Due to the high density, ultra microporous structure, and strong regional limitations of OMS, the static adsorption capacity of MNi-Pyz for NH$_3$ reached 29.1 mmol·g$^{-1}$ at 298 K and 1 bar, which was

more than three times the capacity of industrial standard zeolites (13X Zeolite) and only lower than the current benchmark MOF (LiCl@MIL-53-(OH)$_2$). The adsorption kinetics showed that the materials reached adsorption saturation at low concentration within 20 s, and the maximum adsorption rate was as high as 1.67 mmol·(g·s)$^{-1}$, demonstrating the rapid NH$_3$ capture performance. Additionally, GCMC simulation calculations indicated there were three priority adsorption sites for NH$_3$ with unsaturated metal sites as the primary adsorption sites. Furthermore, the assembled IDE device was used as an NH$_3$ sensor, capable of detecting NH$_3$ in a low concentration range of 1–1000 ppm. These sensors exhibited a low detection limit of 25 ppb and a fast response time of 5 s at room temperature, exhibiting the fastest response speed among all reported electrical signal sensing materials at room temperature. Excellently, the sensor can be recovered by simply purging nitrogen at room temperature, and the response signal can stably output for at least two months at room temperature. In addition, MNi-Pyz-sensors exhibited excellent selectivity towards NH$_3$ in the presence of 13 other potential interfering gases. Prominently, this work provides important guidance for the design and manufacture of high-performance sensors operating at room temperature.

## Methods

### General
Unless otherwise stated, all materials were commercially available and used without further purification. Ni(NO$_3$)$_2$·6H$_2$O, Co(NO$_3$)$_2$·6H$_2$O and K$_2$[Ni(CN)$_4$] were purchased from Aladdin (98% purity), pyrazine was purchased from Macklin (98% purity).

### Synthesis of MNi-Pyz
The synthesis method of MNi-Pyz is fine-tuned on the basis of literature reports[49]. M(NO$_3$)$_2$·6H$_2$O (Ni: 0.872 g, Co: 0.873 g, 3 mmol) was dissolved in a mixture of methanol (15 mL) and deionized water (15 mL). Pyrazine (0.240 g, 3 mmol) was dissolved in a mixture of methanol (15 mL) and deionized water (15 mL). When they were completely dissolved, the two solutions were combined and stirred. Subsequently, K$_2$[Ni(CN)$_4$] (0.723 g, 3 mmol) was dissolved in deionized water (5 mL), and dropped into the above mixture with constant stirring. After 8 min, the reactant was centrifuged to obtain the corresponding purple and pink powder samples for NiNi-Pyz and CoNi-Pyz, respectively. The materials were washed twice with deionized water and methanol, and then dried in a vacuum drying oven until use.

### Characterization and test analysis
**X-ray powder diffraction.** In this experiment, Rigaku Altima IV powder diffractometer from Japanese Neo Confucianism was used for relevant tests. The diffractometer parameters were set as follows: 40 kV,

40 mA, CuK α 1,2 λ = 1.5418 Å. The measurement parameters included scanning speed 5 (°) min$^{-1}$, scanning step 0.02 (°) and scanning range 5 (°)–40 (°). For Temperature-dependent PXRD, the measured parameter included a scan speed of 5 ° min$^{-1}$, a step size of 0.02 (°) and a scan range of 2θ from 5 (°) to 40 (°).

**77K N$_2$ adsorption-desorption experiment.** The specific surface area and pore size of materials are usually characterized by N$_2$ adsorption-desorption at 77 K. The test was measured using ASAP 2020 adsorption equipment, BET and DFT model were used to evaluate the specific surface area and pore size. The specific experimental method was as follows: approximately 100 mg of MNi-Pyz samples were placed on the activation station at 150 °C for 10 h, and then transferred to the analysis port for analysis.

**SEM analysis.** A Hitachi SU3500 SEM instrument was used for acquiring particle morphology images using a 30 kV energy source under a vacuum

**Thermogravimetric analysis.** TGA was conducted on Netzsch STA449F3 thermal analyzer from room temperature to 800 °C (5 K·min$^{-1}$) in air atmosphere using crucible.

**Stability tests of MNi-Pyz.** For solvent stability, as-synthesized samples, about 20 mg for each batch, were immersed in 10 mL of an aqueous solution of pH = 1 (HCl), pH = 13 (NaOH), and boiled water for one day, respectively. Further, prepare saturated KCl (RH = 85%) solution, put the saturated solution and the vial containing MNi-Pyz into a large beaker, seal the beaker with a preservative film and place it at room temperature. The treated MNi-Pyz samples were washed with methanol several times and dried at room temperature before PXRD measurements.

For NH$_3$ stability, calculate the volume of the seal pipe, put MNi-Pyz into the seal pipe, extract and empty the gas inside. Then, according to the definition of parts per million (ppm) a certain volume proportion of NH$_3$ is injected into the sealed pipe, sealed tightly, and placed at room temperature to study their gas stability.

**NH$_3$ adsorption isotherm.** In this experiment, we used the BSD-PMC1 adsorption instrument of Beishide Instrument Company. The specific method is as follows: using MNi-Pyz (150–250 mg) to test the single component gas adsorption curve, the test temperatures are 273 K and 298 K.

**Dynamic test.** The time-dependent adsorption curves of MNi-Pyz on NH$_3$ were measured on the BSD-PMC adsorption instrument. About 50 mg of NiNi-Pyz and CoNi-Pyz were first loaded to the sample chamber and activated at 150 °C under high vacuum for 8 h. After cooling to a specific temperature, 1 bar of NH$_3$ is introduced into the chamber. With the progress of material adsorption, the NH$_3$ pressure change every second is recorded, so as to judge the mass of the sample loaded with gas molecules. As the completion of NH$_3$ consumption in the chamber, after the instrument is balanced for a period of time, a certain amount of gas will be automatically introduced for continuous test recording.

**Molecular simulation.** In order to better understand the adsorption behavior of MNi-Pyz on NH$_3$, Grand Canonical Monte Carlo (GCMC) simulation was carried out on Materials Studio 2019 version. Considering the rigidity of the frame, the forcite module was used to optimize the geometry of MNi-Pyz and NH$_3$. Under the condition of ultra-fine mass, the universal force field (UFF) is used to model it to reach the minimum energy. The partial charge of atoms in the frame is determined by QEq method, detailed structural information can be found in Supplementary Data 1. The simulations were carried out at 298 K, adopting the locate task, Metropolis method in Sorption module and the universal force field (UFF). The Coulomb potential and Leonard Jones 6–12 (LJ) potential are used to calculate the interaction energy between hydrocarbon molecules and frameworks. The cut-off radius of LJ potential is selected as 12.5°, and the long-range electrostatic interaction is treated by Ewald & Group summation method. The load step and balance step are $1 \times 10^5$, and the production step is $1 \times 10^6$. The calculation formula of the binding energy between the skeleton and gas molecule is: $\Delta E = E(MOF) + E(gas)\text{-}E(MOF + gas)$.

**Gas sensing device and method.** Use self-built equipment to test the sensing signal. The gas sensor test by using HIOKI LCR-3536 full-automatic measurement system, and the clamp is L2000-four terminal Kelvin clamp. To reduce the disturbance of noise, insert the noise filter into the power line during testing. Before each test, perform open circuit compensation and short circuit compensation on the instrument, ranging from 4 Hz to 8 MHz, to reduce testing errors and select LCR for measurement mode. The measurement frequency is set to 8 kHz, the test signal level V is set to 5 V, the limit value is adjusted to OFF; select internal trigger mode, set the range to AuTo (1 kΩ-1 MΩ), Low Z selects OFF mode, measurement speed is MED, and the average number of times is three; trigger delay of 0.000 s; DC bias setting OFF, DC adjustment function setting On, JUDGE SYN setting ON.

Preparation of sensors: First, ultrasonic clean the interdigital electrode with deionized water, ethanol and acetone successively for 10 min, and then dry it with nitrogen. The prepared MNi-Pyz is uniformly dispersed in methanol solution with a concentration of about 5 mg·mL$^{-1}$. After, about 200 μL suspension droplets were applied to the electrode and dried naturally in air to form a film. Then the coating sensor is placed in the self-made testing room, and the material is connected to the LCR instrument with a clamp to detect the change of electrical signal, the sensor unit is installed on the sensor equipment through two probes. Before the test, the sample is first activated under vacuum at 80 °C for 2 h (this activation process only needs to be done once before the first use of the sensor), then the test chamber is purged with pure nitrogen to remove the substances left in the chamber until the baseline is balanced. The subsequent material activation steps can be achieved only by blowing with pure nitrogen gas. When the chamber is filled with nitrogen atmosphere, the chamber exhaust valve is closed to keep it in a closed environment, and then a certain volume of NH$_3$ is injected through a micro syringe to test the NH$_3$ sensing performance. To control the tested gas/steam concentration, we calculated the required gas volume in detail according to the definition of the fixed volume of the chamber (220 cm$^3$), ppm (parts per million concentration) and the Antoine equation[50–53]. For Antoine formula, the constants A, B and C for the measured gases are given in Supplementary Table 6. The formula is as follows:

$$\lg P = A - B/(t + C) \tag{1}$$

where $P$ is the vapor pressure of the substance in mmHg; $t$ is the temperature in °C. Equation (1) applies to most compounds, while for some other substances that require only the values of the constants $B$ and $C$, the Eq. (2) can be used to calculate.

$$\lg P = -52.23B/(t + C) \tag{2}$$

The temperature is maintained at 298 K through the temperature console during the test. After the test is completed, precise gas sensing measurement can be achieved by simply blowing nitrogen gas for regeneration. The gas sensing response is defined by Eq. (3):

$$S = (R_a - R_0)/R_0 = \Delta R/R_0 \tag{3}$$

where $R_0$ and $Ra$ represent the electrical impedance of the sensing material under nitrogen atmosphere and target gas respectively. Response and recovery times are marked as the time to reach 90% of the total signal change.

**LOD calculation.** The theoretical limits of detection (LOD) were calculated using reported protocols[30,54]. First, the root mean squared (rms) value − representing the noise-based deviation in $\Delta R/R_0$ − was calculated using the baseline trace before exposure to analyte. We took 600–1000 consecutive points (N) and fit the data to a polynomial. We then calculated sum of squared residuals (SSR) from Eq. (4). The root-mean-square deviation (RMS) was then calculated by Eq. (5). Then plotted concentration of analyte versus response ($\Delta R/R_0$) after a specific exposure time and isolated the range of values wherein this relationship was linear. Linear regression provided an equation of best-fit (slope = m). With these values, we extrapolated the theoretical LOD from Eq. (6).

$$SSR = \sum(y_i - y)^2 \tag{4}$$

$$RMS = (SSR/N)^{1/2} \tag{5}$$

$$LOD = 3 \times RMS \cdot m^{-1} \tag{6}$$

where, where $y_i$ is measured $\Delta R/R_0$ and $y$ is the value calculated from the polynomial fit.

## Data availability
All data supporting the findings of this study are available within the article, as well as the Supplementary Information file, or available from the corresponding authors on request.

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

## Acknowledgements

The authors acknowledge the National Natural Science Foundation of China (21971126), 111 projects (B12015), Nankai University Large scale Instrument Experiment Technology Research and Development Project (23NKSYJS04), and Frontiers Science Center for New Organic Matter (63181206). We also thank Professor Yafei Li's team from Nanjing Normal University for the structural simulations of MOFs.We also thank Professor Heping Ma from Xi'an Jiaotong University for testing the ammonia sorption data.

## Author contributions

Z.Z. conceived and designed the project. S.W. and Y.F. performed the experiments and S.W. wrote the manuscript. T.W., W.L., J.W., and P.Z. helped to analyze the results. H.M. helped to test the adsorption of MOFs. Y.C. and C.P. give important comments on writing.

## Competing interests

The authors declare no competing interests.
