## [Peer Review File · Nature Communications]

Reviewers' Comments:

Reviewer #1:

Remarks to the Author:

This manuscript describes two Hoffman-type MOFs utilized for gas sensing at room temperature with high sensitivity and selectivity due to the high adsorption capacity and fast capture dynamics. The adsorption mechanism is detailedly unveiled not only by dynamic adsorption but also by theoretical studies. Besides, the manuscript is well-written and well-organized, the sensing performance outstands compared with the reported studies. Given all these aforementioned factors, I, thus, would be happy to recommend this manuscript for publication in Nature Communication after minor revision. Hereafter lists the detailed comments.

1. The two MOFs mentioned within this manuscript seem conductive, what is the conductivity mechanism of the two MOFs?
2. It is highly recommended to give FESEM images of the MOF coatings regarding surface morphology and thickness of the films studied for gas sensing.
3. The sensitivity of the sensors seems decreased under higher RH, what might be the reason?
4. Using the template-directed approach and/or the solution processability of MOFs might afford MOF films with great homogeneity and controllable thickness (ACS Appl. Mater. Interfaces 2020, 12, 36715; Adv. Mater. 2021, 33, 2101257). The authors may consider these two approaches in their further studies.

Reviewer #2:

Remarks to the Author:

This manuscript describes fabrication of robust and cost-effective Hoffman-type MOFs and their application as an ammonia sensor. By simply googling the material itself, $\text{Co}(\text{pyz})[\text{Ni}(\text{CN})_4]$ has been already reported as a porous material for highly selective separation of acetylene. So the material itself is not new. This should be clearly declared in the introduction. For ammonia sensing experiments, core application with the proposed materials, critical information is missing. First initial resistances of the coated MOFs will depend on the thickness of material and quality of coating. Simple drop casting method with nanoparticle-like MOFs usually doesn't form a uniform coating on IDE chips. The authors need to show the reproducibility. Second, the measurement setup is not well described. The resistance measurements were done with an LCR meter. Which AC frequency was used and why? How each gas/vapour concentration is controlled for selectivity tests? Why those gas/vapours are chosen? Was there any other gas which potentially has high affinity to the proposed MOFs based on your DFT and GCMC? Third, humidity needs to be clearly mentioned when the LOD is determined because the humidity will decrease the performance and change the LOD. All of these comments need to be addressed before accepting this manuscript for publication.

Reviewer #3:

Remarks to the Author:

Prof. Zhang et al have prepared a solid article about room temperature NH_3 sensing properties of MOF sensors. The description of the preparation of the samples is thoroughly presented, and the structural characterization of the samples is backed up by a large amount of different data. Also, the gas sensing performance results of the samples are clearly shown. The stability and adsorbance/desorption properties of the materials studied seem to be very good.

However, to my opinion, there are some comments and concerns still present in this paper. These concerns are more related to the gas sensing methodology and the gas sensing behaviour of the samples, rather than for the actual sample manufacturing and composition:

Comment 1. XRD results in the supplementary material:

- a.) In Figs 13 a) and 14 a), there are shifts in the main diffraction peaks 2θ angles to lower values

after the NH₃ exposure starts.

b.) In Fig. 13 b), there is a left shoulder appearing in the diffraction peaks around 12.5 degrees and 17.5 degrees in the 500 ppm and 1000 ppm exposure.

c.) In Fig. 14 b), there is a left shoulder appearing in the diffraction peaks around 12.5 degrees and 17.5 degrees after the NH₃ exposure. Also, some shifting of 2θ angles is seen.

Why are these changes described above taking place after the NH₃ exposure? Chemical reactions with the NH₃ gas affecting the crystal structure of the samples? These changes in the XRD results should be commented on the manuscript.

Comment 2. Gas sensing methodology (Figures 6 and 7 + supplementary material):

a.) How long are the gas pulses used in the NH₃ sensing measurements? This is not described in the manuscript. It is impossible to see any saturation in the gas sensing electrical pulses at the moment. Saturation of the electrical signal is needed in practical applications, so that the sensors can detect the amount of gases in the atmosphere if needed.

b.) Are all the gas sensing results measured only in pure N₂ gas background? The tests should be also made with synthetic air background. I would assume that most practical applications used in RT are present in air atmosphere, not pure N₂ gas. The gas sensing properties, including the effect of interfering gases to selectivity of the sensing material, might be highly different in air atmosphere than in N₂ background.

c.) It was mentioned in the manuscript, that before the gas sensing measurements, the MOF materia had to be activated in vacuum at 80C. How often this activation has to be made; e.g. only before the 1st gas sensing measurement, and how long is this "activation" valid ? Thinking again about practical applications, this might be quite tricky...

d.) In the manuscript, there was a nice comparison between the MOF materials presented in this paper, and other materials studied. However, if I understood correctly, there was no comparison of the sensitivity between different materials. For example, at 1000 ppm of NH₃ sensitivity value of $S = 0.007$ with the formula $S = (R_a - R_0)/R_0$ means basically that for example, when $R_0 = 100 \Omega$, the result will be $R_a = 100.7 \Omega$. Meaning the actual impedance change would be 0.7Ω in the presence of 1000 ppm of NH₃. This seems a very low increase of value compared to state-of-art gas sensing results. And again in practical applications, it would be tricky to use LCR-meter to detect gases in the atmosphere; for these low relative changes in the electrical signal you cannot use very simple measurement devices in a reliable way.

All the comments made for the gas sensing methodology above should be carefully addressed, as they are also the main results of the manuscript.

Reviewer #1:

This manuscript describes two Hoffman-type MOFs utilized for gas sensing at room temperature with high sensitivity and selectivity due to the high adsorption capacity and fast capture dynamics. The adsorption mechanism is detailly unveiled not only by dynamic adsorption but also by theoretical studies. Besides, the manuscript is well-written and well-organized, the sensing performance outstands compared with the reported studies. Given all these aforementioned factors, I, thus, would be happy to recommend this manuscript for publication in Nature Communication after minor revision. Hereafter lists the detailed comments.

Response: We thank the reviewer for the high comments and support of our work.

Comments 1: The two MOFs mentioned within this manuscript seem conductive, what is the conductivity mechanism of the two MOFs?

Response: We thank the reviewer for the comment. According to the suggestion, we have tested the conductivity (about 10^{-7} S/cm), indicating the MOFs belonging to the semiconductor (consistent with the results reported in Adv. Mater. 2017, 29, 1605071). This is because π electrons can be delocalized and excited from electron-rich, conjugated π orbital ligands (HOMO) to unoccupied d orbitals (LUMO) in transition metals.

Comments 2: It is highly recommended to give FESEM images of the MOF coatings regarding surface morphology and thickness of the films studied for gas sensing.

Response: We thank the reviewer for this suggestion. Per your suggestion, the FESEM images of coatings were added, and the thickness was tested. The results are listed in Supplementary Fig. 20. and highlighted in yellow.

Comments 3: The sensitivity of the sensors seems decreased under higher RH, what might be the reason?

Response: We thank the reviewer for the suggestion. The sensors can be used normally below RH 60%, which is able to meet the application requirements. The decrease in sensitivity under high humidity conditions can be attributed to the following factors:

(1) Impact of humidity on MOF adsorption capacity: Supplementary Fig. 8 shows a sharp increase in the water adsorbed at around 60%RH. In higher humidity, water molecules can block the pores and compete with the target gas in the MOFs leading to decreased sensor sensitivity.

(2) Effect of humidity on the sensor interface: High humidity can cause the formation of a water film on the sensor surface, altering the interface properties between the sensor and the gas. This change in interface characteristics can weaken the interaction between the sensor and the target gas, thereby reducing sensitivity (*Chem. Soc. Rev.* 2014, 43, 5594-5617; *Microporous Mesoporous Mater.* 2018, 265.).

(3) Electrochemical effects: Humidity can influence the electrochemical reactions occurring at the sensor surface. It may modify the charge distribution or electron transfer characteristics, affecting the overall sensitivity of the sensor (*J. Am. Chem. Soc.* 2019, 141, 2046-2053; *ACS Sens.* 2017, 2, 1294-1301.).

In response to this valuable feedback, we have added related descriptions and explanations in the revised manuscript (highlighted in yellow).

Comments 4: Using the template-directed approach and/or the solution processability of MOFs might afford MOF films with great homogeneity and controllable thickness (*ACS Appl. Mater. Interfaces* 2020, 12, 36715; *Adv. Mater.* 2021, 33, 2101257). The authors may consider these two approaches in their further studies.

Response: We thank the reviewer for this suggestion. We sincerely appreciate your thorough review of our paper and your valuable suggestion. Your feedback has provided us with a new perspective and direction for our future material research, and we are grateful for the opportunity to further explore these methods.

Reviewer #2:

This manuscript describes fabrication of robust and cost-effective Hoffman-type MOFs and their application as an ammonia sensor. By simply googling the material itself, Co(py_z)[Ni(CN)₄] has been already reported as a porous material for highly selective separation of acetylene. So, the material itself is not new. This should be clearly declared in the introduction.

Response: We thank the reviewer for this suggestion. Per your suggestion, we added a brief description of this material in the introduction (highlighted in yellow) to ensure that readers have a thorough understanding of the material's background. At the same time, we clarified the differences and contributions of our research. It should be noted that the reported application of acetylene separation is completely different from the NH₃ sensing application in our manuscript. Thus, it doesn't affect the novelty of this work.

Comments 1: For ammonia sensing experiments, core application with the proposed materials, critical information is missing. First initial resistances of the coated MOFs will depend on the thickness of material and quality of coating. Simple drop casting method with nanoparticle-like MOFs usually doesn't form a uniform coating on IDE chips. The authors need to show the reproducibility.

Response: We thank the reviewer for the comment. For your suggestion, we supplement the reproducibility and reliability verification for NH₃ sensing experiments in Figure R1. The excellent reproducibility of this method was demonstrated by five independent experiments. Further, this method and the stability of the prepared material have also been well verified in literature (*J. Am. Chem. Soc.* 2019, 141, 2046-2053; *Angew. Chem. Int. Ed.* 2023, e202302996; *Adv. Funct. Mater.* 2020, 30, 2006598).

Figure R1. Five independent experiments were repeated using (a) NiNi-Pyz and (b) CoNi-Pyz sensors to detect NH_3 in different ppm concentration ranges (1 to 1000 ppm).

Comments 2: Second, the measurement setup is not well described. The resistance measurements were done with an LCR meter. Which AC frequency was used and why? How each gas/vapour concentration is controlled for selectivity tests? Why those gas/vapours are chosen? Was there any other gas which potentially has high affinity to the proposed MOFs based on your DFT and GCMC?

Response: Thank you very much for the questions. The measurement conditions were added in the main text (highlighted in yellow).

(1) When it comes to the selection of AC frequency, we considered the following factors: we chose a relatively low frequency (as 6 kHz~9 kHz) to ensure that the conductivity of the coating MOFs is not affected by charge transfer dynamics. Low frequency can eliminate the influence of charge accumulation and polarization effects in the medium, thereby providing more accurate response measurement results. We found in the experiment that high sensitivity can be achieved in the frequency range of 6 kHz to 9 kHz. In addition, a lower frequency (< 6 kHz) may cause interference from capacitive effects. In comparison, a high (>9 kHz) frequency may reduce the test's sensitivity, while the instrument accuracy may also decrease. The settings of other parameters are also supplemented and highlighted in the main text.

Figure R2. Detection of NH₃ in different ranges of ppm concentrations (1~1000 ppm) at different frequencies using (a~c) NiNi-Pyz and (d~f) CoNi-Pyz sensors.

(2) To control the tested gas/steam concentration, we calculated the required gas volume in detail according to the definition of the fixed volume of the chamber (220 cm³), ppm (Parts per million concentration) and the Antoine equation. Then, a micro injector was used to inject the corresponding volume of gas/steam into the chamber. This experimental design can help us evaluate the performance of sensors in different gases and concentrations, and provide analysis of sensor selectivity. In the revised draft, we will provide a more detailed explanation of our methods for controlling the concentration of gases/vapors. We have provided a detailed description of the operation process and added it to the main text (highlighted in yellow).

(3) In our study, we mainly focused on investigating the adsorption of NH₃ using MNi-Pyz. This is because, in the context of our research on environmental monitoring, NH₃ is the gas we want to study, so studying the affinity of NH₃ is of great significance for practical applications. The specificity and selectivity of materials are also critical, and then several interfering gases are selected for testing. The gas/steam we choose is usually related to NH₃ sensor application of the coating MOFs, such as

some typical volatile organic compounds and reducing gases (*J. Mater. Chem. A.* 2021, 9, 4150-4158; *J. Mater. Chem. A.* 2015, 3, 1174-1181; *Adv. Funct. Mater.* 2020, 30, 1909756; *Angew. Chem. Int. Ed.* 2017, 56, 16510-16514). In the revised text, we have provided explanations for selecting these gases/vapors (highlighted in yellow).

(4) Due to the presence of unsaturated metal sites and ultramicroporous structure in the material, we recognize that other gases may have a certain affinity for the proposed MOFs in sensor research. We have added and highlighted the calculation results of other gases (such as common gas components in the air and some VOC gases) to provide a more comprehensive research perspective in the main text and supporting information (Supplementary Fig. 13 and 14).

Comments 3: Third, humidity needs to be clearly mentioned when the LOD is determined because the humidity will decrease the performance and change the LOD. All of these comments need to be addressed before accepting this manuscript for publication.

Response: We thank the reviewer for this comment. Our LOD determination is in pure nitrogen condition without additional increasing humidity. Per the suggestion, we have also added LOD values for other humidity conditions, and specific details have been added to the method in the main text and supporting information (highlighted in yellow).

Reviewer #3:

Prof. Zhang et al have prepared a solid article about room temperature NH₃ sensing properties of MOF sensors. The description of the preparation of the samples is thoroughly presented, and the structural characterization of the samples is backed up by a large amount of different data. Also, the gas sensing performance results of the samples are clearly shown. The stability and adsorbance/desorption properties of the materials studied seem to be very good.

However, to my opinion, there are some comments and concerns still present in this paper. These concerns are more related to the gas sensing methodology and the gas sensing behaviour of the samples, rather than for the actual sample manufacturing and composition:

Response: We thank the reviewer for the high comment and support of our work.

Per your comments and concerns about the gas sensing performance and methodology, we have added a more detailed description and nuanced analysis in the main text and SI to make the gas sensing part clearer (highlighted in yellow).

Comments 1: XRD results in the supplementary material:

a.) In Figs 13 a) and 14 a), there are shifts in the main diffraction peaks 2θ angles to lower values after the NH_3 exposure starts.

b.) In Fig. 13 b), there is a left shoulder appearing in the diffraction peaks around 12.5 degrees and 17.5 degrees in the 500 ppm and 1000 ppm exposure.

c.) In Fig. 14 b), there is a left shoulder appearing in the diffraction peaks around 12.5 degrees and 17.5 degrees after the NH_3 exposure. Also, some shifting of 2θ angles is seen.

Why are these changes described above taking place after the NH_3 exposure? Chemical reactions with the NH_3 gas affecting the crystal structure of the samples? These changes in the XRD results should be commented on the manuscript.

Response: We thank the reviewer for the comment and suggestion. We carefully analyzed our PXRD data and found the issue could be because our instrument encountered problems with the light tube's aging and its lifespan's expiration, which may lead to peak splitting and deviation. We have replaced the PXRD instrument's light tube and retested the PXRD patterns in the revised version (Supplementary Fig. 15 and 16). The new results showed no changes in the crystal structure.

Supplementary Fig. 15. PXRD patterns of (a) NiNi-Pyz, (b) CoNi-Pyz after placed at 100 ~ 1000 ppm NH_3 for one day.

Supplementary Fig. 16. PXRD patterns of (a) NiNi-Pyz, (b) CoNi-Pyz after placed at 1000 ppm NH_3 for a week to a month.

Comments 2: Gas sensing methodology (Figures 6 and 7 + supplementary material):

a.) How long are the gas pulses used in the NH_3 sensing measurements? This is not described in the manuscript. It is impossible to see any saturation in the gas sensing electrical pulses at the moment. Saturation of the electrical signal is needed in practical applications, so that the sensors can detect the amount of gases in the atmosphere if needed.

Response: We thank the reviewer for the comment. In practical operation, we first fill the chamber with nitrogen and seal it tightly. Then, based on the gas concentration to be detected and the total volume of the chamber being 220 cm³, calculate the required amount of gas and gradually inject the gas into the chamber. During this process, there is no specific pulse duration. On the contrary, we determine the concentration of NH₃ by measuring the changes in the concentration of the injected gas. After gas injection into the cavity, equilibrium can be quickly achieved, and there is a positive correlation between response and concentration. We have provided a detailed description of the operation process and highlighted it in the main text.

After a detailed analysis of the high-concentration test results, we found that when the gas concentration reached 3500~4000 ppm, the signal line ultimately reached the saturation state in Figure R3.

Figure R3. Detection of NH₃ in different ranges of ppm concentrations (1~4000 ppm) on (a) NiNi-Pyz and (d) CoNi-Pyz sensor.

b.) Are all the gas sensing results measured only in pure N₂ gas background? The tests should be also made with synthetic air background. I would assume that most practical applications used in RT are present in air atmosphere, not pure N₂ gas. The gas sensing properties, including the effect of interfering gases to selectivity of the sensing material, might be highly different in air atmosphere than in N₂ background.

Response: We thank the reviewer for the comment. In the revised draft, we explore the signals of our sensor components under air background and discuss the possibilities for use in real-world environments.

From the results, it can be seen that the sensors have similar response behavior in air atmosphere (RT= 298 K, RH≈ 35~55%) with the nitrogen atmosphere. Due to the interference of gases, suspended particles, and temperature and humidity in the air, the signal slightly decreases. However, it still has good cyclic stability and excellent selectivity, which can be further used in practical work environments. In the revised manuscript and SI (listed in Supplementary Fig. 23 and 24), we have highlighted the related discussion in yellow.

c.) It was mentioned in the manuscript, that before the gas sensing measurements, the MOF material had to be activated in vacuum at 80C. How often this activation has to be made; e.g. only before the 1st gas sensing measurement, and how long is this "activation" valid? Thinking again about practical applications, this might be quite tricky...

Response: We thank the reviewer for the question. This "activation" process only needs to be done once before the 1st gas sensing measurement. This is because solvent molecules occupy channels when MOFs are taken from the mother liquor. After activation, the pores and functional sites are fully exposed, which can be used stably without further reactivation. We found that the activated MOF material could maintain its sensing behavior for more than half a year. During this time, the device could be regenerated by simply blowing N₂. Therefore, the operation is very convenient and close to practical applications. Detailed operating instructions have been added in the main text (highlighted in yellow).

d). In the manuscript, there was a nice comparison between the MOF materials presented in this paper, and other materials studied. However, if I understood correctly, there was no comparison of the sensitivity between different materials. For example, at 1000 ppm of NH₃ sensitivity value of $S = 0.007$ with the formula $S = (R_a - R_0)/R_0$ means basically that for example, when $R_0 = 100 \Omega$, the result will be $R_a = 100.7 \Omega$. Meaning the actual impedance

change would be 0.7Ω in the presence of 1000 ppm of NH_3 . This seems a very low increase of value compared to state-of-art gas sensing results. And again in practical applications, it would be tricky to use LCR-meter to detect gases in the atmosphere; for these low relative changes in the electrical signal you cannot use very simple measurement devices in a reliable way.

Response: We thank the reviewer for this suggestion. Compared to other conductive materials, NiNi-Pyz and CoNi-Pyz MOFs have a higher basic resistance value and a lower rate of change with the same amount of change. But this change can be actually observed and is related to the concentration of ammonia gas. A similar phenomenon has been reported in earlier studies (*Nat. Commun.* 2019, 10, 1328; *Adv. Mater.* 2016, 28, 5229; *Angew. Chem. Int. Ed.* 2019, 58, 14089; *Angew. Chem. Int. Ed.* 2016, 55, 15879; *J. Mater. Chem. A*, 2018, 6, 5550; *ACS Nano* 2017, 11, 9276).

Regarding the use of LCR testing (*Adv. Mater.* 2019, 31, 1807161), the sensitivity of LCR detection can be improved by adopting the instrument circuit design, adding signal amplifiers, or the adoption of precise calibration methods in the future. Although there are some challenges in gas detection, LCR instruments hold significant value for real-time data visualization, monitoring, alarms, security detection, and other scenarios. Regarding the use of MNi-Pyz materials, in our future research, we will further enhance material performance and make it more suitable for practical applications. For example, we will explore material composites or cascades with other materials to optimize their sensing capabilities, offering better results to the application of our materials in the field of sensors, and improving their practicality and reliability.

Overall, although our research is currently in the laboratory stage, the study of the material's performance on NH_3 demonstrates the promising potential of MNi-Pyz for sensor applications in practical conditions and lays a solid experimental foundation for future integration with real-world applications. Herein, we want to emphasize our main contribution one more time. (1) The MOF materials possess ultra-high adsorption capacity for NH_3 , which is only lower than the

current benchmark material (LiCl@MIL-53-(OH)₂). (2) The assembled IDE device can detect NH₃ in a low concentration range (1~1000 ppm) with a low detection limit of 25 ppb and a fast response time of 5 s. It exhibits the fastest response speed among all reported NH₃ sensors (output electrical signals) at room temperature. (3) Our sensors can effectively avoid the impact of oxidation or pollution on cycle stability and can be restored by simply blowing nitrogen gas at room temperature (which greatly reduces operating costs compared to sensors that require additional heating). (4) We fully utilized various technologies, such as static adsorption behavior, kinetic data, and GCMC simulation, to gain deeper insights into the material's sensing mechanism, which cannot be achieved by traditional sensing materials.

Your valuable suggestions have greatly inspired our work, and we express our sincere appreciation.

Again, we thank the reviewers for the constructive comments and suggestions, which have made our manuscript much improved.

Thank you very much for your favorable consideration of our manuscript.

Sincerely,

Zhenjie Zhang, Professor of Chemistry

Reviewers' Comments:

Reviewer #1:

Remarks to the Author:

The authors have fully addressed my comments. So, I support its publication in Nature Communications.

Reviewer #2:

Remarks to the Author:

The authors well addressed most of my previous comments.

However, my first comment was about chip to chip variation, not repeatability with one chip. If the authors deposit the MOF with five different chips. How much variation of initial resistance/impedance is going to be? After normalization of sensing signal, how much variation will we see? Maybe not much but I would like to see the data. In addition, the authors did comment on what the initial resistance/impedance was.

Other than this some minor typos like Hoffman vs Hofmann, PH vs pH, NH₃ vs NH₃ etc. need to be corrected in the final version.

Reviewer #3:

Remarks to the Author:

The response authors have provided is at adequate level. Therefore I recommend the paper for publication.

Reviewer #1:

The authors have fully addressed my comments. So, I support its publication in Nature Communications.

Response: We thank the reviewer for the support of our work.

Reviewer #2:

The authors well addressed most of my previous comments.

Response: We thank the reviewer for the high comment and support of our work.

Comments 1: However, my first comment was about chip to chip variation, not repeatability with one chip. If the authors deposit the MOF with five different chips. How much variation of initial resistance/impedance is going to be? After normalization of sensing signal, how much variation will we see? Maybe not much but I would like to see the data. In addition, the authors did comment on what the initial resistance/impedance was.

Response: We thank the reviewer for this valuable suggestion. Upon introducing five distinct chips into the MOF framework, we noted variations in initial resistance, with NiNi-Pyz and CoNi-Pyz MOF-IDEs exhibiting changes within the ranges of $637,019 \pm 18 \Omega$ and $689,016 \pm 13 \Omega$, respectively. The corresponding initial data has been graphed in Figure R1. When detecting concentrations ranging from 1 ppm to 1000 ppm, we observed a change in resistance between 230 ~ 5000 Ω and 135 ~ 4800 Ω for NiNi-Pyz and CoNi-Pyz MOF-IDEs, respectively. This change can be actually observed during testing and is related to changes in ammonia concentration. The following figure shows the sensing performance of five different chips.

Figure R1. Five independent experiments were repeated using (a) NiNi-Pyz and (b) CoNi-Pyz sensors to detect NH₃ in different ppm concentration ranges (1 to 1000 ppm).

Comments 2: Other than this some minor typos like Hoffman vs Hofmann, PH vs pH, NH₃ vs NH₃ etc. need to be corrected in the final version.

Response: We thank the reviewer for this suggestion. We have checked the text carefully and revised all mistakes.

Reviewer #3:

The response authors have provided is at adequate level. Therefore I recommend the paper for publication.

Response: We thank the reviewer for the support of our work.

Reviewers' Comments:

Reviewer #2:

Remarks to the Author:

The authors addressed all the comments I made so I recommend for publication.

Reviewer #1:

The authors have fully addressed my comments. So, I support its publication in Nature Communications.

Response: We thank the reviewer for the support of our work.

Reviewer #2:

The authors addressed all the comments I made so I recommend for publication.

Response: We thank the reviewer for the support of our work.

Reviewer #3:

The response authors have provided is at adequate level. Therefore I recommend the paper for publication.

Response: We thank the reviewer for the support of our work.